# l-Buthionine Sulfoximine Detection and Quantification in Polyurea Dendrimer Nanoformulations

**DOI:** 10.3390/molecules24173111

**Published:** 2019-08-27

**Authors:** Pedro Mota, Rita F. Pires, Jacinta Serpa, Vasco D. B. Bonifácio

**Affiliations:** 1CQFM-IN and IBB—Institute for Bioengineering and Biosciences, Instituto Superior Técnico, Universidade de Lisboa, 1049-001 Lisboa, Portugal; 2CEDOC, Chronic Diseases Research Centre, NOVA Medical School, Faculdade de Ciências Médicas, Universidade NOVA de Lisboa, Campo dos Mártires da Pátria 130, 1169-056 Lisboa, Portugal; 3Instituto Português de Oncologia de Lisboa Francisco Gentil (IPOLFG), Rua Prof. Lima Basto, 1099-023 Lisboa, Portugal

**Keywords:** buthionine sulfoximine, polyurea dendrimers, catechol, nanoformulation, UV-Vis detection, derivatization

## Abstract

l-Buthionine sulfoximine (l-BSO) is an adjuvant drug that is reported to increase the sensitivity of cancer cells to neoplastic agents. Dendrimers are exceptional drug delivery systems and l-BSO nanoformulations are envisaged as potential chemotherapeutics. The absorption of l-BSO at a low wavelength limits its detection by conventional analytical tools. A simple and sensitive method for l-BSO detection and quantification is now reported. In this study, l-BSO was encapsulated in a folate-targeted generation four polyurea dendrimer (PURE_G4_-FA_2_) and its release profile was followed for 24 h at pH 7.4 and 37 °C. The protocol uses in situ l-BSO derivatization, by the formation of a catechol-derived *orto*-quinone, followed by visible detection of the derivative at 503 nm. The structure of the studied l-BSO derivative was assessed by NMR spectroscopy.

## 1. Introduction

The detection of organic molecules that absorb at low wavelengths is a critical issue and a challenge for analytical chemistry. Ultraviolet radiation having wavelengths less than 200 nm is difficult to handle and is rarely used as a routine tool for structural analysis. To overcome this problem, chemical derivatization is often employed in order to conduct indirect spectrophotometric analysis. Therefore, depending on the reactivity and available functional groups of the analytes, we can choose from a variety of chemical or enzymatic reactions in order to bring absorption to longer wavelengths [1].

Buthionine sulfoximine (BSO) is a specific γ-glutamylcysteine synthetase (γ-GCS) inhibitor that blocks a rate-limiting step in glutathione (GSH) biosynthesis by inhibiting the synthesis of glutamylcysteine, the first precursor of GSH [2]. BSO is a chiral molecule and its racemate was first successfully synthesized in 1979 [3]. The two isomers (D and L) were also prepared and investigated (Figure 1), however l-buthionine-(*S,R*)-sulfoximine, contrary to the racemate and the D isomer, shows a high inhibition efficacy [4]. More recently, a new improved synthetic route of the racemate, using mild and safe conditions, was also reported [5].

The fact that BSO is capable of altering metabolism makes it a promising key molecule for cancer therapeutics, and its role in cell growth inhibition, apoptosis induction [6], drug resistance reduction [7], and as a cancer adjuvant therapeutic in ovarian clear cell carcinoma [8], and others [9] has already been investigated.

Nevertheless, BSO detection and quantification in biological assays is still limited, mainly due to its very low molar absorptivity and an absorption maximum below 200 nm [10]. Therefore, its detection is difficult using high-performance liquid chromatography (HPLC) using a UV detector. BSO detection, along with isomers separation, was first reported in 1987 using HPLC after sample derivatization with *o*-phthalaldehyde [11]. However, this method has several limitations. Because of the instability of the *o*-phthalaldehyde derivative at room temperature, each sample was prepared immediately prior to injection onto the HPLC column. Additionally, butyrophenone, used as the internal standard, required a fluorescence detector in addition to the UV detector. Another limitation of this method was the prolonged time of the analysis (>75 min/sample). Later, in 1993, an alternative method was developed. In this case, derivatization was performed using phenylisothiocyanate, which produces a more stable derivative [12], but its high toxicity is a major drawback.

In this work, we developed a simple, safe, and fast methodology for visible detection of l-Buthionine sulfoximine (l-BSO) using a folate-targeted polyurea dendrimer nanoformulation.

## 2. Results and Discussion

### 2.1. l-BSO Encapsulation and Release Studies

Aiming at future in vivo studies, the l-BSO release was performed at 37 °C under physiological conditions (pH 7.4). However, due to the stability of the reagents and derivatives at lower pHs, the method is also applicable in studies involving tumor cells or tissues under acidic environments [13]. PURE dendrimers were chosen as a model drug delivery system due to their reported biocompatibility and biodegradability [14]. It is well known that PURE dendrimers in aqueous solutions have the ability to produce positively charged species [15] as a result of amines protonation. This feature leads to a structure expansion as a result of charge repulsions between the dendrimer branches, thus speeding cargo release. A polyurea generation four dendrimer, targeted with folate (PURE_G4_-FA_2_) (Figure 2), was synthesised and loaded with l-BSO. Folate targeting is aimed towards therapeutic applications in ovarian clear cell carcinoma.

Figure 3 shows the l-BSO release profile from a l-BSO@PURE_G4_-FA_2_ nanoformulation at pH 7.4. The release was followed for 24 h and the results indicate a burst release in the first hours, as expected in this type of delivery system [16,17]. After 1 h, around 60% of the loaded drug was released to the medium, reaching a plateau after 3 h. After 24 h, 90% of l-BSO was released, meaning that only a residual amount of l-BSO was trapped inside the PURE_G4_-FA_2_ dendrimer during this period of time. The quantification of BSO released from the PURE_G4_-FA_2_ dendrimer was determined using a calibration curve, obtained from an assay under the same experimental conditions (see Appendix A). In more acidic media, a faster release is expected.

### 2.2. l-BSO Detection and Quantification

As stated before, the detection and quantification of BSO is problematic due to its very low molar absorptivity and absorption wavelength (<200 nm). To overcome this issue, we developed a new detection protocol using catechol as the derivatizing agent (Scheme 1).

It is known from literature that ortho phenols (e.g., catechol) react with sodium periodate, a strong oxidizing agent, to give the corresponding o-quinones [18]. The reaction is fast and the formed o-quinones are highly reactive coloured intermediates (absorption bands ca. 390 nm), which easily undergo nucleophilic attack [19]. Therefore, we investigated this reaction in the presence of l-BSO in order to evaluate its applicability in a detection and quantification protocol. Since we found (data not shown) that the derivatization reaction is time dependent, thus affecting the absorbance intensity, all the triplicate data points were rigorously collected at the same time (60 s) using a chronometer. In this derivatization reaction, a colour change from yellow (o-quinone) to red (l-BSO derivative, 503 nm) was observed.

In order to clarify the derivatization mechanism, the reaction was followed by ^1^H NMR (Figure 4).

In this experiment, sodium periodate was added to a NMR tube containing a solution of catechol in D_2_O (Figure 4A). The formation of the corresponding o-quinone is visualised by a colour change to yellow (Figure 4B). Then, a solution of BSO in D_2_O was added and spectra were recorded at intervals of 10 min until no more changes in the spectra were observed (total time 50 min) (Figure 4D). The spectra show the formation of the Michael adduct, as expected. The ^13^C-NMR spectrum (see Appendix A) was recorded and a downshift was observed, especially in the region 49–54 ppm. Also, the spectrum did not show any signals around 180-190 ppm [20], which corroborates the formation of the BSO derivative **3**. The higher downshift was observed for the carbon at 53.2 ppm, and the carbon linked to the BSO primary amine [5]. From these data, we may conclude that the primary amine of BSO reacts with o-quinone, thus originating intermediate **3** as a major product.

This strategy is envisaged as a simple and straightforward methodology for the detection and quantification of analytes with low wavelength absorption and having a nucleophilic functional group.

## 3. Materials and Methods

### 3.1. Reagents and Materials

l-Buthionine-(*S*,*R*)-sulfoximine (l-BSO) (≥97% purity) was obtained from Sigma-Aldrich. All chemicals and solvents were used as received without further purification. Polyurea (PURE) dendrimer generation four (PURE_G4_) was synthesized following our reported supercritical-assisted polymerization [21].

### 3.2. Synthesis of PURE_G4_-FA_2_

Folate-targeted polyurea dendrimer generation four (PURE_G4_-FA_2_, Figure 2) was prepared by reacting PURE_G4_ with activated folic acid succinic ester (FA-NHS) (Scheme 2). 

FA-NHS was synthesized following the literature [22]. Typically, in a round bottom flask, 250.0 mg (0.566 mmol) of folic acid (FA) was dissolved in dimethylsulfoxide (DMSO) (2.75 mL). After the addition of 130.8 mg (1.137 mmol) of *N*-hydroxysuccinimide (NHS), 128.5 mg (0.623 mmol) of dicyclocarbodiimide (DCC), and 0.15 mL (1.082 mmol) of triethylamine (TEA), the reaction was stirred at room temperature (RT) overnight in the dark. The product was precipitated and washed several times with diethyl ether. After drying under vacuum, FA-NHS was obtained as a yellow powder (263.4 mg) in 86.4% yield. ^1^H NMR (400 MHz, DMSO-*d6*) δ (ppm): 8.64 (1H, s), 7.63 (2H, d, *J* = 8.0 Hz), 6.64 (2H, d, *J* = 8.0 Hz), 4.49 (2H, s), 4.28 (1H, s), 2.54 (4H, s), 2.29 (1H, s), 2.03 (1H, s), 1.93 (1H, s).

Next, FA-NHS was conjugated with PURE_G4_ (via NH_2_ surface groups) to obtain PURE_G4_-FA_2_. In a 25 mL bottom round flask, 100 mg (0.0127 mmol) of PURE_G4_ was dissolved in 5.0 mL of DMSO. To this solution, 13.68 mg (0.0254 mmol) of FA-NHS and 6.9 μL (0.051 mmol) of TEA were added. The reaction was stirred at RT overnight in the dark. Next, TEA excess was removed on the rotary evaporator and diethyl ether was added. The obtained precipitate was dried under vacuum and PURE_G4_-FA_2_ was obtained as yellow oil in 93.9% yield. By NMR, it was found that two molecules of folic acid had been conjugated to the surface of PURE_G4_. ^1^H NMR (400 MHz, D_2_O) δ (ppm): 8.64 (2H, s), 7.70 (4H, bs), 6.86 (4H, d, *J* = 8.0 Hz), 4.61 (2H, s), 3.54–3.00 (180H, m), 2.96–2.40 (462H, m).

### 3.3. Encapsulation of l-BSO in PURE_G4_-FA_2_

In a vial, 78.0 mg (8.90 µmol) of PURE_G4_-FA_2_ was dissolved in 20 mL of ethanol. Then, 21.6 mg (0.0972 mmol) of l-BSO was added and the mixture was vigorously stirred. The encapsulation occurred at RT overnight, in the dark, for 24 h. Afterwards, no l-BSO on suspension was observed and the product was purified by dialysis for 30 min (MWCO 100–500 Da). After evaporation of the solution, the product was dried under vacuum and characterized by ^1^H NMR. The amount of l-BSO loaded into the dendrimer (BSO@PURE_G4_-FA_2_) was determined by ^1^H NMR. ^1^H NMR (300 MHz, D_2_O) δ (ppm): 8.60 (s, 2H), 7.63 (br, 4H), 6.78 (br, 4H), 3.62 (s, 16H), 3.44–3.07 (m, 180H), 3.06–2.40 (m, 462H), 1.76 (m, 32H), 1.45 (q, *J* = 6.0 Hz, 30H), 0.92 (t, *J* = 6.0 Hz, 48H).

### 3.4. l-BSO Release Profile

l-BSO release studies were performed at 37 °C in sodium phosphate buffer medium (PBS, pH 7.4). First, 6.3 mg of l-BSO@PURE_G4_-FA_2_ were dispersed in 1 mL of medium and placed in a SnakeSkin™ dialysis membrane (MWCO 3500 Da). The dialysis bag was then immersed in 60 mL of release medium and kept in at constant temperature with stirring. Samples (1 mL) were periodically collected and replaced by the same volume of fresh medium. The amount of l-BSO released was determined by UV-Vis spectroscopy. The release of l-BSO from the dendrimers was obtained in triplicate.

### 3.5. Quantification of BSO by UV-Vis Spectroscopy

Using a modified protocol [18], l-BSO was derivatized in order to be detected by UV-Vis spectroscopy. l-BSO quantification was performed by adding to the samples 300 µL of catechol 2.25 mM in PBS, followed by 300 µL of sodium periodate 6.75 mM in PBS. After 60 sec, the absorption of the l-BSO derivative (503 nm) was measured in a PerkinElmer Lambda 25 UV-Vis Spectrometer with a slit width of 5 nm at a scan rate of 240 nm min^−1^ at 25 °C. For the calibration curve, standard solutions of l-BSO were prepared in the concentration range of 0.1–150 µM in PBS and processed using the same protocol. A good correlation coefficient (R^2^ = 0.997) was obtained (see Appendix A).

## 4. Conclusions

In summary, the detection and quantification of l-buthionine sulfoximine was performed using a polyurea dendrimer nanoformulation. The protocol is fast, accurate, simple, non-toxic, and may be applied to detect and quantify other similar drugs.

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
