# Peer review of "l-Buthionine Sulfoximine Detection and Quantification in Polyurea Dendrimer Nanoformulations"

_molecules, 2019, doi:10.3390/molecules24173111_

Round 1

Reviewer 1 Report

                                     Comments

This manuscript described a simple method to detect the BSO and applied it to the quantitation of BSO released from folate-targeted polyurea dendrimer nanoformulations. Using the catechol-derived strategy, BSO is conjugated with catechol to form a derivative which has an absorption of 503nm. This is a promising method for BSO assay in biochemical studies. In my opinion, this manuscript can be accepted for publishing in Molecules after minor revision. But the following issues must be addressed carefully.

Some questions and suggestions are listed as follows:

In the section of “3.2. Synthesis of PUREG4-FA2”, the authors should give a schematic diagram of the related chemical reactions. Recently, BSO was used as an adjuvant to improve the antitumor effects of chemotherapeutic agents. In the section of “introduction (line 44-46, Page 2)”, I suggest the authors should add these references, such as “Fan et al., Cancers, 2019, 11, 317”. folate-target or “folate-targeted”, this should be consistent in your manuscript. In the section of “2. Results and Discussion”, the authors should first introduce the detection method, mechanism, followed by the practical application for the detection and quantitation of BSO in folate-targeted polyurea dendrimer nanoformulations. In Figure 4, why the chemical shifts of -OH and -COOH were not presented? Some grammatical errors should be revised, such as “Is well known that PURE……”, “Is known from literature……”, “in order evaluate its applicability……”, “The quantification of L-BSO was performed by adding to the samples 300 µL of catechol 2.25 mM in PBS, followed by 300 µL of sodium periodate 6.75 mM in PBS” and so on.

Author Response

Response to Referee

The manuscript was revised according to the referee’s comments. The new/corrected text was added to the manuscript using a blue colour.

Q1: In the section of “3.2. Synthesis of PUREG4-FA2”, the authors should give a schematic diagram of the related chemical reactions.

R1: A scheme showing the synthetic process was added to the manuscript.

Q2: Recently, BSO was used as an adjuvant to improve the antitumor effects of chemotherapeutic agents. In the section of “introduction (line 44-46, Page 2)”, I suggest the authors should add these references, such as “Fan et al., Cancers, 2019, 11, 317”.

A2: The suggested reference was added to the manuscript.

Q3: folate-target or “folate-targeted”, this should be consistent in your manuscript.

A3: Corrected.

Q4: In the section of “2. Results and Discussion”, the authors should first introduce the detection method, mechanism, followed by the practical application for the detection and quantitation of BSO in folate-targeted polyurea dendrimer nanoformulations.

A4: We appreciate the referee suggestion. However, to us it makes more sense the actual order: synthesis/encapsulation/release, detection/quantification/mechanism. Nevertheless we revised this section and added a missing subtitle (2.2).

Q5: In Figure 4, why the chemical shifts of -OH and -COOH were not presented?

A5: The NMR spectra were recorded in D2O. We believe that this is the reason why these signals are not being observed (solvent exchange).

Q6: Some grammatical errors should be revised, such as “Is well known that PURE……”, “Is known from literature……”, “in order evaluate its applicability……”, “The quantification of L-BSO was performed by adding to the samples 300 µL of catechol 2.25 mM in PBS, followed by 300 µL of sodium periodate 6.75 mM in PBS” and so on.

A6: The authors thank the referee for the pointed errors. The manuscript was revised and errors were corrected.

Reviewer 2 Report

Dear Authors,

Please find my comments below.

The Authors discuss a method for detecting L-buthionine sulfoximine (L-BSO) via its derivatisation and its application for determining the amount of L-BSO released from polyurea dendrimer drug carrier. I believe that the subject matter is original and interesting in terms of both basic and applied research. The contributions presented in the manuscript are significant enough to warrant publication. That said, the manuscript contains a number of statements that I perceive as unclear or insufficiently supported by the presented experimental evidence. In my opinion, there are also some minor factors, mostly related to the presentation of the results that adversely affect the Reader’s comprehension of this interesting manuscript.
As such, I recommend the publication of the manuscript in MDPI Materials only pending a general overhaul of the manuscript, with the most important issues and comments being listed below:

Major remarks:

Introduction:
The Authors are citing literature from 1987 and 1993 without any inclusion of more modern literature. Why is that so? Have there been no further works published that are relevant?

Minor remarks:

I advise altering the abbreviation for polyurea dendrimers, as “PURE” can be easily misunderstood as referring to the purity of the system.

I advise marking the D2O signal in the NMR spectra presented in the manuscript (Figure 4). It would also be beneficial for the individual proton signals to be assigned to the protons present in each of the investigated molecules.

The English used in the manuscript is generally satisfactory, but numerous minor punctuation and grammar mistakes are present; some of those are listed below:

Line 28: “analytic chemistry” - should be “analytical chemistry”
Line 29: “difficult to handle, and” - should be “difficult to handle and”
Lines 29-30: “To overcome this problem chemical derivatization is often a choice to undergo indirect spectrophotometric analysis.” - should be e.g. “To overcome this problem, chemical derivatization is often employed, in order to conduct indirect spectrophotometric analysis.”
Line 44: “Targeting altered metabolism turns BSO a promising key molecule” - this is unclear and I suggest rephrasing, perhaps “The fact that BSO is capable of altering metabolism makes it a promising key molecule”?
Lines 51 and 52: “o-phthaladehyde” - should be “o-phthalaldehyde”
Line 62: “Aiming future in vivo studies” - should be “Aiming at future in vivo studies”
Line 64: “studies involving tumors cells or tissues” - should be either “studies involving tumor cells or tissues” or “studies involving tumorous cells or tissues”
Line 65: “were chosen as model drug delivery system due” - should be “were chosen as a model drug delivery system, due”
Line 67: “produce positively charge species” - should be “produce positively charged species”

Author Response

Response to Referee

The manuscript was revised according to the referee comments. The new/corrected text was added to the manuscript using a blue colour.

Q1: The Authors are citing literature from 1987 and 1993 without any inclusion of more modern literature. Why is that so? Have there been no further works published that are relevant?

A1: We appreciate the referee comment. In fact, to the best of our knowledge, there is no more recent work regarding BSO detection.

Q2: I advise altering the abbreviation for polyurea dendrimers, as “PURE” can be easily misunderstood as referring to the purity of the system.

A2: We appreciate the referee comment. However, the acronym PURE (always using capital letters) has been used for many years regarding this particular class of dendrimers.

Q3: I advise marking the D2O signal in the NMR spectra presented in the manuscript (Figure 4). It would also be beneficial for the individual proton signals to be assigned to the protons present in each of the investigated molecules.

A3: The signal from D2O and the protons from the corresponding molecules were fully assigned in figure 4. A brief stating (ans corresponding reference) supporting our structural proposal for intermediate 3 base on 13C NMR was also added: “Also, the spectrum does not show any signals around 180-190 ppm”.

Q4: The English used in the manuscript is generally satisfactory, but numerous minor punctuation and grammar mistakes are present; some of those are listed below:

Line 28: “analytic chemistry” - should be “analytical chemistry”Line 29: “difficult to handle, and” - should be “difficult to handle and”Lines 29-30: “To overcome this problem chemical derivatization is often a choice to undergo indirect spectrophotometric analysis.” - should be e.g. “To overcome this problem, chemical derivatization is often employed, in order to conduct indirect spectrophotometric analysis.”Line 44: “Targeting altered metabolism turns BSO a promising key molecule” - this is unclear and I suggest rephrasing, perhaps “The fact that BSO is capable of altering metabolism makes it a promising key molecule”?Lines 51 and 52: “o-phthaladehyde” - should be “o-phthalaldehyde”Line 62: “Aiming future in vivo studies” - should be “Aiming at future in vivo studies”
Line 64: “studies involving tumors cells or tissues” - should be either “studies involving tumor cells or tissues” or “studies involving tumorous cells or tissues”
Line 65: “were chosen as model drug delivery system due” - should be “were chosen as a model drug delivery system, due”Line 67: “produce positively charge species” - should be “produce positively charged species”

A4: The authors thank the referee for the pointed errors. The manuscript was revised and errors were corrected.